



# Constraints on average alpha recoil distance during $^{238}$U decay in baddeleyite (ZrO$_2$) from atom probe tomography

Donald W. Davis[1], Steven Denyszyn[2], Denis Fougerouse[3]

[1]Earth Sciences, University of Toronto, Toronto, M5S 3B1, Canada
[2]Earth Sciences, Memorial University of Newfoundland, St. John's, A1A 0G3, Canada
[3]Geoscience atom probe facility, John de Laeter Centre and School of Earth and Planetary Sciences, Curtin University, Perth Australia

*Correspondence to*: Donald W. Davis (dond@es.utoronto.ca)

**Abstract.** Atom probe tomography of $^{238}$U and $^{206}$Pb has been applied to baddeleyite crystals from the Hart Dolerite (1791 $\pm$ 1 Ma) and the Great Dyke of Mauritania (2732 $\pm$ 2 Ma) in an effort to constrain the average nuclear recoil distance of U-series daughter nuclei and thereby correct U-Pb ages determined on small baddeleyite crystals for alpha-recoil loss of Pb. Both crystals were thought to expose natural crystal surfaces providing a boundary where maximum recoil loss could be observed, but both surfaces showed no adjacent variations in Pb concentrations. However, the Great Dyke sample shows U zoning and

the associated $^{206}$Pb zoning is affected by alpha recoil. A forward modelling approach was used where $^{206}$Pb redistribution functions were determined for a range of possible alpha recoil distances and synthetic $^{206}$Pb/$^{238}$U profiles were determined from the convolution of the observed U profile with the redistribution functions. These can be compared to the observed $^{206}$Pb/$^{238}$U profile. A complication is that the 400 nm range of sampling is lower than the range of possible alpha recoil redistribution effects. In order to get a realistic match to the observed $^{206}$Pb/$^{238}$U profile, it was necessary to extrapolate the

observed zoning as an oscillatory pattern. This gives a best estimate for the average alpha recoil distance of about 40 nm.

## 1 Introduction

Baddeleyite (monoclinic ZrO$_2$) is a common geochronometer in silica-undersaturated rocks. Like zircon (tetragonal ZrSiO$_4$), U is incorporated into its crystallizing lattice but not Pb. Unlike zircon, radiation damage from the decay of U and its radioactive

daughters has minimal effect in the form of Pb loss (Rioux et al. 2010; Lumpkin 1999). However, minor discordance (<3%) between the $^{206}$Pb/$^{238}$U and $^{207}$Pb/$^{235}$U systems is common in baddeleyite of all ages, outside of uncertainties in U decay constants (Rioux et al. 2010; Schoene et al. 2006). Apparent daughter loss in younger samples (< ca. 1000 Ma), where age interpretations must be based on $^{206}$Pb/$^{238}$U ratios rather than $^{207}$Pb/$^{206}$Pb ratios, is manifested as younger $^{206}$Pb/$^{238}$U dates but data are still generally concordant due to relatively large $^{207}$Pb/$^{235}$U uncertainties.

Baddeleyite does not appear to be susceptible to low temperature alteration like radiation-damaged zircon, although it can break down to polycrystalline zircon with accompanying Pb loss during metamorphism (Davidson and van Breemen, 1988). This may be because its chemical stability is due to its composition whereas that of zircon is due to its structure, which is



disrupted by radiation damage. Therefore, pre-treatment methods such as air abrasion (Krogh 1982) or annealing and HF-leaching ("chemical abrasion", Mattinson 2005) that are designed to remove alteration and are frequently employed to mitigate

Pb loss in zircon are generally unnecessary (e.g., Rioux et al. 2010). Various mechanisms have been proposed for observed discordance in baddeleyite. They include micro- or cryptocrystalline zircon overgrowths, which are susceptible to extensive Pb loss (e.g., Darling et al. 2016; Schmieder et al. 2015); loss of intermediate daughter $^{222}$Rn through diffusion (Heaman & LeCheminant 2001); excess intermediate daughters $^{231}$Pa (Ivanov et al. 2021; Sun et al. 2020; Amelin & Zaitsev 2002) and $^{230}$Th (Wu et al. 2015); and daughter loss through alpha recoil (Denyszyn et al. 2009; Davis & Sutcliffe 1985). In particular,

apparent preferential $^{206}$Pb loss leading to a biased effect on the $^{206}$Pb/$^{238}$U ratio has been observed with improved precision on single analyses, and has been attributed to excess $^{231}$Pa (Ibañez-Mejia & Tissot, 2019) and/or $^{222}$Rn mobility (Pohlner et al. 2020). An observed relationship between grain size, specifically surface area-to-volume ratio, and degree of $^{206}$Pb loss where crystal rims are more strongly affected than cores, implies either fast-pathway/volume diffusion of $^{222}$Rn (Pohlner et al. 2020) or alpha recoil as a cause. As alpha recoil is a fundamental physical process that must occur with every radioactive decay,

determining average alpha-recoil distance is important to establish the degree to which discordance in baddeleyite can be attributed to this process.

Alpha recoil refers to the displacement of daughter radioisotopes as a result of the ejection of an alpha particle from the parent. The resulting lattice damage to the mineral's crystal structure facilitates Pb loss resulting from low-temperature alteration, a phenomenon well-documented in zircon (e.g., Nasdala et al. 2010). Pb (and intermediate daughters in the U decay

series) can also be directly ejected from the crystal. Recoil distances for single alpha decays have been calculated to range from 20 to 33 nm in zircon (Nasdala et al. 2010), and while the directions of the alpha emission and recoil are random, an average cumulative recoil distance from the 8 decays of the $^{238}$U decay series should be about 3 times the average single recoil distance. There are only 7 decays in the $^{235}$U decay series, which may account for the apparent younging bias in $^{207}$Pb/$^{206}$Pb ages. Therefore, a zone of U-daughter depletion can be expected to make up the outer ca. 50-100 nm of a given zircon crystal

(Davis & Davis 2018), representing a volume in which alpha-recoil loss can be significant. Even in zircon crystals that are relatively equant compared to the flat, bladed habit of baddeleyite, and therefore have relatively low surface area-to-volume ratios, alpha recoil can generate measurable discordance (Romer 2003). For a crystal of 100 μm in longest dimension and 2:1 aspect ratio, this depletion zone can make up ca. 0.3% of the volume of intact crystals which have not undergone physical or chemical abrasion (Schmieder et al. 2015).

Alpha recoil distances in baddeleyite have not been precisely determined. Denyszyn et al. (2009) tested the requisite correlation of grain size and apparent age of baddeleyites on the order of 10 μm thickness, and could not resolve any effect on $^{206}$Pb/$^{238}$U ratios at the scale of typical ID-TIMS U-Pb analyses, suggesting that recoil distances in baddeleyite are shorter than those for zircon. Davis & Davis (2018) used a sensitive high-resolution ion microprobe (SHRIMP II) to create depth profiles of $^{206}$Pb/$^{238}$U ratios away from natural crystal surfaces in baddeleyite. Analytical issues limited the ability to precisely

determine recoil distances with this method, but a value of 24 ± 7 nm was obtained. This distance is similar to that of zircon, and would imply resolvable discordance for crystals of about 15 μm in size. The alpha recoil distance in baddeleyite has



implications for a minimum useful volume of baddeleyite for geochronology, and for the selection of subsamples for spatially-detailed analysis (e.g., FIB-TIMS, White et al. 2020). Accurate knowledge of the recoil distance would allow at least first order corrections to ID-TIMS Pb/U ages based on the dimensions of samples.

Atom probe tomography (APT) can be used to directly observe the distribution of individual atoms of a trace element in a crystal lattice (Reddy et al., 2020). APT has been applied to baddeleyite in previous studies, mostly using the baddeleyite mineral standard Phalaborwa (Reinhard et al. 2018) and/or crystals that underwent shock metamorphism (White et al. 2019; White et al. 2017). One germane finding of these studies is that while major elements are homogeneously distributed at the atomic level in baddeleyite, radiogenic Pb and incompatible trace elements (Si, Mg, Al, Yb and Fe) can be heterogeneously

distributed in nanoscale domains (White et al 2017; White et al 2018). The nanoscale U distribution in Phalaborwa baddeleyite crystals was also shown to be variable with localized zonation, and heating baddeleyite to temperatures up to 500°C does not seem to affect that distribution (White et al 2017). We apply APT to two baddeleyite crystals of known age and construct profiles of U and radiogenic Pb distribution in order to constrain average alpha recoil distance in baddeleyite for the $^{238}$U decay chain.

**2 Methods**

**2.1 Sample preparation**

Samples were selected based on identification of well-defined crystal faces, concordant U-Pb analyses, and relatively old age and high U content to ensure measurable abundances of U and radiogenic Pb. Baddeleyites were chosen from the Hart Dolerite (M2, $^{207}$Pb/$^{206}$Pb age of 1790.5 ± 1.4 Ma; Ramsay et al., 2019, their sample GS11043-1) and the Great Dyke of Mauritania

(M5, upper-intercept age of 2733 ± 2 Ma; Tait et al., 2013, their sample GTD-8).

Both baddeleyite grains were picked and placed on individual scanning electron microscope (SEM) aluminium stubs covered by carbon tape (Fig. 1A and B). The stubs were sputter-coated with an approximately 200 nm layer of Cr, to serve both as a conductive layer and as a cap protecting the surface of the baddeleyite grains. Atom probe specimens were prepared from the Cr-coated grain with a Ga$^+$ Tescan Lyra3 Focused Ion Beam (FIB) at the Microscopy and Microanalysis Facility

(MMF), Curtin University. A Pt layer was sublimated along the surface of the crystal in order to protect the region of interest from Ga implantation and Pt sublimation was used to fuse the needle to the APT specimen holder. The FIB was operated at an accelerating voltage of 30 kV during the sculpting of the needle-shaped specimens and at 2 kV in the final stage to remove the external layer affected by the high-energy Ga beam. During the final stage of polishing, some Cr cap was intentionally left at the apex of the specimens to ensure stable evaporation of the baddeleyite grain surface during atom probe analyses, and to

identify the original crystal surfaces (Fig. 2A and B).

**2.2 Atom Probe Tomography (APT)**

APT relies on the field evaporation of ions from a needle-shaped specimen. The original position of the ions in the analysed volume is given in three dimensions and at sub-nm resolution by a position-sensitive detector. The ions are identified by their





time-of-flight and reported on a mass-to-charge ratio spectrum (mass spectrum). This study used the Geoscience Atom Probe (Cameca LEAP 4000X HR), at Curtin University. The instrument was operated in laser assisted mode with a UV ($\lambda = 355$ nm) laser set at 100 pJ pulse energy and at a repetition rate of 125 kHz. The specimens were maintained at 50 K base temperature in order to inhibit surface diffusion during analysis. In the mass-to-charge ratio spectra, peaks higher than twice the background were identified and ranged for 3 dimensional reconstructions using Cameca's APsuite 6.3 software. Voltage evolution

reconstructions were performed using a detector efficiency of 0.36, an image compression factor of 1.65 and a k-factor of 3.3. For baddeleyite, the atomic volume was calculated at 0.01133 nm$^3$/atom and the electric field was empirically determined at 29.08 V/nm (Fougerouse et al., 2022). Tomographic data for one specimen of each grain were successfully acquired, with 62 million ions for specimen M5 (Great Dyke of Mauritania) and 65 million ions for specimen M2 (Hart Dolerite). Depth concentration profiles were generated for a 5 nm bin size and exclude the Cr cap.

The U and Pb isotopic compositions were quantified from the atom probe data by using a narrow range (0.1 Da) for each isotopic ionic specie. Uranium was spread over several ionic species with the dominant $^{238}UO_2^{++}$ (135 Da), but also $^{238}UO_2^+$ (270 Da) and $^{238}UO^{+++}$ (84.7 Da). No peaks were visible above background for $^{235}U$. Lead was present as $^{206}Pb^{++}$ and $^{207}Pb^{++}$ (103 and 103.5 Da, respectively). The local background signal was measured by selecting a wide 'peak free' region (0.5 to 1 Da) in proximity to each ionic specie and normalized to the width of the specie range (0.1 Da). Uncertainties were estimated

at 1σ using counting statistics and propagating background corrections adapting protocols previously defined for baddeleyite, zircon and monazite (Fougerouse et al. 2018; Peterman et al. 2016; White et al. 2017).

## 3 Results and Discussion of analyses and modelling

### 3.1 Constraints on alpha recoil distance from U zoning

The U concentration in specimen M5 (Mauritania) is higher than in specimen M2 (Hart) with an average U of 288 ppma (parts per million atomic) for M5 and 23 ppma for M2. In specimen M5, the distribution of U is heterogeneous with a gradient of concentration from low U content near the surface and higher content towards the centre of the grain (Fig. 3). The U concentration ranges from approximately 150 ppma to 650 ppma.

The effect of alpha recoil is to randomly redistribute a fraction of the U atoms, resulting in a distribution of radiogenic Pb

atoms that differs from the U distribution in having smaller concentration gradients. The degree of broadening can be calculated for an assumed average alpha recoil distance and compared to the measured profile, thus providing a way to potentially constrain its value. The largest U gradient should be encountered at the crystal surface. The Cr cap coating preserved at the tips of the atom probe specimens confirms the assumed position of the crystal surfaces (Fig. 2). Surprisingly, profiles of background-corrected abundances of Pb and U for each crystal (Fig. 3) show no depletion of Pb in the outermost 10-50 nm.

For specimen M5, the abundances of these elements is non-uniform, with a zone of relative enrichment in the 200-400 nm depth range, which is interpreted to represent growth zoning of U. The sampled surfaces of the baddeleyite crystals appeared to be natural crystal surfaces based on visual inspection under SEM imaging and the absence of obvious fracturing. In the



Davis and Davis (2018) study, one out of the three studied grains also did not show decreasing Pb near the crystal boundary and it was concluded that the boundary may have been a cleavage plane, which may also be the case here.

Sample M2 shows a uniform U concentration (Fig. 3) and is of no use for constraining alpha recoil distance. However, the step in U concentration along the axis of sample M5 (Fig. 3) should result in Pb concentration effects dependent on alpha recoil distance. M5 also has more radiogenic Pb because it is the oldest and shows the highest U concentration, which makes parent/daughter ratios easier to measure. The measured $^{206}$Pb/$^{238}$U ratio profile in M5 is shown in Fig 4. The measured age of M5 requires an equilibrium radiogenic $^{206}$Pb/$^{238}$U ratio of 0.53. The measured $^{206}$Pb/$^{238}$U profile shown in Fig. 4B is notably

higher than this for distances less than 200 nm, corresponding to the low U zone and slightly lower for greater distances corresponding to the high-U zone. If the recoil distance were very small (less than a few nanometres), Pb would not be significantly displaced from U and one would expect the $^{206}$Pb/$^{238}$U profile to be constant, independent of the U concentration, which is not the case. If the recoil distance were very large, Pb would be redistributed approximately independent of the U profile and should have a constant composition. In this case, the ratio profile would be proportional to the U concentration.

The measured profile does show that the $^{206}$Pb/$^{238}$U ratios are low where the U concentration is high and high where it is low but the U concentrations differ by a factor of over 3 whereas the ratio levels differ by about 1.5. Therefore, alpha recoil must be limited to a few tens of nanometres, which would result in excess Pb in the low U zone from recoils originating in the high-U zone, which itself should show a deficiency of radiogenic Pb. The profile in Fig 4B generally accords with this since the $^{206}$Pb/$^{238}$U ratio is notably higher than 0.53 in the range 0-150 nm and slightly lower in the range above 200 nm.

Modelling of the $^{206}$Pb concentration was carried out based on the observed $^{238}$U distribution and assumed values of R, the average value of the recoil distance for each of the 8 alpha emitting nuclides. This was done by calculating an estimation of the redistribution function for a given R value and calculating the convolution of this function with the observed U distribution. As in Davis and Davis (2018), the planar symmetry of a zoned U distribution simplifies the calculations, allowing the results to be expressed as 2 dimensional profiles where the redistribution function varies only with distance along the axis, normal to

the plane of zoning.

An average recoil distance is used for simplicity in formulating the problem but it does not describe the true kinetics of a recoiling nucleus. Each of the 8 nuclei in the $^{238}$U chain decay with different energies and so will have different average recoil distances. Also, recoiled nuclei transfer energy to host atoms by multiple scattering events, which is the source of radiation damage, so their actual paths are complicated and cover a range of values even for the same decay. None of this is relevant to

the problem of correcting $^{206}$Pb/$^{238}$U ages, which requires knowledge of the dimensions of the analysed crystals and the $^{206}$Pb redistribution curve, a function of the average total displacement of a $^{206}$Pb daughter atom from its decayed $^{238}$U nucleus. Thus, even if the R value is an average, it is useful for characterizing their overall displacement.

To approximate the redistribution function for a given alpha recoil distance we start with a uniform plane of atoms and calculate the displacement of an atom normal to the plane after being displaced a distance R in a random direction. This is

done on a population of 1000000 atoms and the shape of the resultant curve of number of atoms versus normal distance, when normalized to an area of 1, approximates the redistribution function for the distance R (Fig. 5A). Calculations of redistributed



Pb profiles were done using Visual Basic for Applications (VBA) in Excel. See Suppl. Data File 1 for an example using an R value of 40 microns. This contains software and instructions for repeating the calculations with different values of R.

The distribution functions after 8 recoils approximate a Gaussian curve (Fig. 5B), which may seem surprising since, as shown in Suppl. Data File 1, the result of 1 recoil is a parabolic-like function and even 4 recoils produce a function that is noticeably different from a Gaussian. Convergence to a Gaussian shape by recursion of any random process, even one whose distribution is itself non-Gaussian, is predicted by the well-known Central Limit Theorem (https://en.wikipedia.org/wiki/Central_limit_theorem).

Alpha recoil results in spreading the decayed U atoms within a given distance bin into adjacent bins, which involves calculating a convolution of the observed U profile with the above determined area-normalized redistribution function. The result gives a theoretical recoiled $^{206}$Pb distribution and profile of $^{206}$Pb/$^{238}$U ratios with the assumed value of R. Note from Fig. 5 that the mean total displacement of a Pb atom is much greater than the R value because there are 8 decays.

Fig. 6 and Suppl. Data File 1 illustrate the result of eight 40 nm (R) alpha recoils for a hypothetical Gaussian-shaped U concentration of similar scale to the peak from the sample. For simplicity, it is assumed that the amount of $^{238}$U at present is equal to the amount of $^{206}$Pb produced. Convolution of a Gaussian curve produces a wider Gaussian curve. The $^{206}$Pb/$^{238}$U profile shows roughly symmetrical peaks on each side of the U peak where excess $^{206}$Pb has been projected from adjacent high U levels on each side of the peak onto part of the tail, raising the ratio from 1, while within the U peak the ratios are less than 1. The measured U profile from APT consists of a portion of a peak and its leftward tail (Fig. 4A). It can be seen that the measured ratio profile (Fig 4B) also consists of parts that have lower ratios over the peak and higher near the tail. The measured ratios over the low U zone on the left are noticeably above the radiogenic equilibrium value of 0.53 and then drop to a constant value slightly but distinctly below the equilibrium value over the high U zone (Fig 4B), but the measured profile shows much flatter regions of high and low ratios than the example in Fig. 6.

The measured APT $^{238}$U profile does not fit a Gaussian curve well since a Gaussian with the same height and half-width tends to tail off more gradually. Also the measured U profile is not wide enough to avoid effects of recoil beyond its measured range. It is therefore necessary to extrapolate the profile above and below. Given that there are two unknowns, one of which is an array, the determination of R will not be unique but it may be possible to constrain it by reasonable assumptions and examining how different U distributions affect the shape of the ratio profile.

Assuming that the U concentrations at <0 nm and >430 nm are constant and have the values of their closest measured points, results in the $^{206}$Pb/$^{238}$U profiles shown in Figure 7, where they overlie the measured profile, for assumed R values of 20 nm, 30 nm and 40 nm (see Suppl. Data File 2). All three models show a deficiency of Pb compared to the observed profile in the first 100 nm and an excess beyond about 350 nm. Their ratios do not deviate from the equilibrium ratio of 0.53 as much as the measured profile. The 20 nm value gives the flattest pattern, which is expected since small values of R approach the equilibrium ratio. The 40 nm value gives the closest fit except near the end of the profile where it deviates more than results from the other values. The deficiency of Pb at low distances suggests that actual U concentrations were higher than the uniform



projection, which was of a lower-U zone, while the excess of Pb at the high distance end suggests that the projection of the high-U zone should have been downward.

        Projecting the end of the observed profile linearly downward and using R = 40 nm (see Suppl. Data File 2, Flat2 sheet) gives a good fit to the right part of the observed profile but the modelled profile is lower than the observed one for low distance values (Figure 8). This suggests that the U concentration was increasing at negative distances, which in turn suggests the
possibility of oscillatory zoning of the U concentration. This should not be unexpected since oscillatory zoning is common, at least in zircon.

        The high U peak and its tail were each approximated by parabolas which were used to interpolate above and below the measured range with an oscillation cycle wavelength of about 600 nm (Figure 9). The convolutions of this profile with redistribution functions for 30 nm, 40 nm and 50 nm are compared to the measured profile in Figure 10 (see Suppl. Data File
2, Parabolic sheets). The assumption of oscillatory zoning resolves the deficiency in Pb at low distance as well as the excess at high distance but there is a dip in the modelled Pb at about 90 nm for all the profiles. This might be present in the measured profile but the increased noise over this low-U range makes it difficult to discern. The effect of increasing the assumed recoil distance is generally to raise the $^{206}$Pb/$^{238}$U ratios in the low-U zone and lower them in the high-U zone at greater distance. The 40 nm value of R gives a remarkably good fit to the data, matching the average ratios over low and high distance ranges. The
agreement is similar if the unmeasured sections of the right side of the parabolas are approximated by a straight line so they decrease more slowly (see Suppl. Data File 2). Increasing the width of the low-U section results in a higher value of R, but drops the early part of the profile as in the flat extrapolation (Figure 7). Thus, the assumption of oscillatory zoning gives ratio profiles that approximately match the shape of the observed profile and seem to require an average recoil distance of about 40 nm. It is possible to improve the fits for 30 nm and 50 nm by varying the widths and heights of zones above and below the
measured section but difficult to match all features of the observed profile. 40 nm seems to match the detailed profile best with the fewest assumptions. R values below 30 nm produce profiles that are too flat to match the data. Because of the lack of knowledge of the U distribution beyond the range of measurement, it is not possible to make realistic error estimations. The reported errors on the data produce unrealistically high values of MSWD, which vary only slightly between models. Errors are therefore probably underestimated and are not as sensitive to correlations (patterns) of data points as a visual fit.

The most robust estimate of about 40 nm is significantly higher than the average recoil distance of 24 ±7 nm found from depth profiling of natural surfaces of baddeleyite crystals using SHRIMP although it is near the upper range of measured values in this work (Davis and Davis, 2018). These measurements were difficult because of the strong dependence of Pb/U biases on the condition of the sputtered surface in secondary ion mass spectrometry. As the hole deepens, the bias must be corrected by comparison with results from analysis of a polished crystal surface. U concentrations were also found to drop off near natural
surfaces during some analyses, which was unexpected, but the average recoil distance was found to be about the same whether or not this was taken as an artefact.



### 3.2 Constraints on alpha recoil distance from U clustering

Throughout specimen M5, U forms hundreds of clusters ~10 nm diameter spaced approximately 20 nm apart. The clusters were quantified and averaged (Fig. 11) using the proximity histogram method of Hellman et al. (2000). "Clusters" is used to describe the regions of relatively high concentration of the atoms of interest; "matrix" refers to the parts of the crystal between the clusters. On average, they are enriched in U (2,406 ppma), Pb (305 ppma), Ti (1,614 ppma) and Fe (2,400 ppma) compared to the "matrix" (U = 225 ppma; Pb = 145 ppma; Ti = 712 ppm; Fe = 2,086 ppma). Although enriched within the U clusters, the Pb distribution is more diffuse than U and the clumping less evident. The Hf composition is unchanged between the clustered or matrix domains. The U clusters are observed in all parts of the dataset, even at U concentrations as low as 150 ppma (Fig. 11). No clusters were observed in specimen M2, in which U and Pb were homogeneously distributed.

For specimen M5, the entire specimen yields a $^{206}Pb/^{238}U$ ratio of 0.528 ± 0.011 for a calculated age of 2,734 ± 70 Ma and a $^{207}Pb/^{206}Pb$ ratio of 0.179 ± 0.017 for an age of 2,640 ± 162 Ma. The matrix domain of the sample yields a $^{206}Pb/^{238}U$ ratio of 0.772 ± 0.020 for a calculated age of 3,687 ± 128 Ma and a $^{207}Pb/^{206}Pb$ ratio of 0.165 ± 0.019 for an age of 2,507 ± 193 Ma. The combined clusters composition is 0.168 ± 0.007 for the $^{206}Pb/^{238}U$ ratio and a $^{206}Pb/^{238}U$ age of 1,002 ± 46 Ma and a $^{207}Pb/^{206}Pb$ ratio of 0.232 ± 0.042 for a $^{207}Pb/^{206}Pb$ age of 3067 ± 292 Ma. In this dataset, the $^{207}Pb^{++}$ peak suffered a high background due to the thermal tail of the $^{206}Pb^{++}$ peak, detrimental for the precise quantification of $^{207}Pb$. The Pb content of specimen M2 was not high enough to calculate meaningful $^{206}Pb/^{238}U$ and $^{207}Pb/^{206}Pb$ ratios.

The mechanism by which these clusters were initially formed is unknown, particularly in light of the homogeneous distribution of U in the Hart Dolerite specimen analyzed in the same run, and in previously reported descriptions of APT carried out on Phalaborwa baddeleyite (Reinhard et al. 2018; White et al. 2018). Several mechanisms have been proposed for the formation of clusters in minerals, including: annealing of radiation damage (Peterman et al., 2021; Verberne et al., 2020); phase exsolution during cooling (Fougerouse et al., 2018); deformation (White et al., 2018; Fougerouse et al., 2019); fluid alteration (Joseph et al., 2023); and during growth (Fougerouse et al., 2016). The samples studied were not subjected to significant metamorphism (above greenschist facies) since crystallization (Ramsay et al., 2019; Tait et al., 2013). The composition of the clusters in specimen M5 differs by approximately 2,500 ppma (0.25 at.%) from the composition of the matrix. This difference in composition indicates that the clusters are not a separate phase (inclusion), but a baddeleyite domain enriched in trace elements. Therefore, the phase exsolution model is unlikely to be responsible for our observations (Fougerouse et al., 2018). Dislocations in minerals are typically observable in atom probe tomography data in the form of trace element enriched linear features (Piazolo et al., 2016). No linear features were observed in specimen M5 and the evidence for crystal-plastic deformation is therefore weak, making it unlikely that the clusters formed as a result of deformation. Baddeleyite is relatively resilient to fluid alteration and clusters are unlikely to have formed through this process. Uranium can substitute to Zr in baddeleyite, however the mismatch in ionic sizes of $Zr^{4+}$ (0.084 nm) and $U^{4+}$ (0.10 nm) limits the range of solid solution and likely results in distortion of the crystal lattice (Kulkarni et al., 2009). During growth, distorted crystal domains enriched in U may be more suitable for the sorption of additional U ions. The incorporation of U in baddeleyite would result in local



depletion of U in the silicate melt at the crystal-melt boundary. The nanoscale U distribution in baddeleyite may therefore be limited by the diffusion of U to the crystal boundary and may self-organise in the pattern observed in our data, as suggested
for nanoscale oscillatory zoning (Putnis et al., 1992; Wu et al., 2019). Therefore, we conclude that the U clusters are primary, formed during initial crystallization of the baddeleyite.

The distribution of Pb should reflect the effect of alpha recoil from the clustered U, generating more diffuse clusters of Pb around a U cluster. In this case, convolution is unnecessary as the excess U on the cluster is effectively a point source with the only relevant parameter for recoiled Pb atoms being the distance from the cluster. However, the relative volumes of spherical
shells around the cluster increase as the cube of their radius, which means that recoiled $^{206}$Pb atoms are rapidly diluted by Pb in the matrix as their recoil distance becomes larger.

The averaged measured $^{238}$U and $^{206}$Pb concentrations and the $^{206}$Pb/$^{238}$U ratios within and outside the clusters are shown on Fig. 12. Distance zero represents the location of the cluster boundary and distance is negative within the boundary. The volume and number of atoms are rapidly reduced as the centre of the cluster is approached, so measurement errors become
large (Fig. 11). For the purpose of modelling, the average cluster is considered to have a radius of 3.5 nm, the radial bins are taken as 0.5 nm wide and the U and Pb concentrations for the first 3 bins are taken as being constant and set at the measured values for the third bin (Fig. 12). Any measurements deeper than this are too imprecise to be meaningful.

Although there is an increase in $^{206}$Pb concentration within the cluster this is well below the amount that would give a $^{206}$Pb/$^{238}$U equilibrium ratio corresponding to the age of the sample (about 0.5). As mentioned above, the $^{206}$Pb/$^{238}$U ratio in the
interior of the cluster is about 0.17. This might be explained if the clusters formed at about 1 Ga and the recoil distance were very small (<1 nm) but there is no reason to expect U mobility at low temperatures, since it is considered to have a blocking temperature even higher than Pb, at least in zircon (Cherniak and Watson, 2003), or that the recoil distance is negligible (see above). There is also a narrow peak in the radial $^{206}$Pb/$^{238}$U ratio at about 5 nm, where it rises above 1 due to the fact that the U concentration decreases near the average cluster boundary. Away from the average cluster the background U and Pb
concentrations reflect the equilibrium $^{206}$Pb/$^{238}$U ratio for the 2.7 Ga age of the sample.

The small size of the clusters makes the Pb/U ratio profile very sensitive to recoils. The modelling shows that average recoil distances above 1 nm effectively remove all radiogenic Pb from the cluster and dilute it into the background, although R distances up to a few nm tend to pile recoiled Pb into the U trough, increasing the peak in the $^{206}$Pb/$^{238}$U profile slightly (Suppl. Data File 3). Therefore the cluster distributions can only be used to constrain the average recoil distance as being greater than
a few nanometers. The presence of the small $^{206}$Pb peak within the average cluster, as well as the trough in U concentration around it cannot be explained realistically by alpha recoil. They are most likely to be primary effects of cluster formation and the depletion of the silicate melt in proximity to the preferential sorption of the U ions in the clusters. Although the clusters are primarily composed of ZrO$_2$, they show elevated concentrations of Fe and Ti, as well as U. The $^{206}$Pb peak might correspond to elevated concentrations of common Pb, which might be confirmed with $^{204}$Pb measurements if this mass could be effectively
resolved. The trough in U concentration is perhaps due to freezing in of a U concentration gradient in the magma following growth of the high-U clusters.





## 4 Corrections to baddeleyite age calculations

Exact calculation of the proportion of recoiled Pb ejected from a baddeleyite crystal of arbitrary shape would require
convolving the U distribution with a spherical function having a Gaussian radial distribution of width corresponding to the average recoil distribution. Since this is a 3-dimensional problem, the number of data points is increased as the cube of those for the 1-dimensional problem treated above, requiring a much longer computation time. For the case of a tabular crystal where the (001) face is much larger than all the others, an approximate correction can be easily determined from the pattern of recoiled Pb near an infinitely large face (Fig 13). The crystal surface will retain 50% of its Pb and the retention factor will increase to
100% inward following the integral of the redistribution function, which approximates the error function of its closest Gaussian curve. For R = 40 nm, 99% of the Pb is retained in the bin at 200 nm and 85% of the Pb in the volume below 200 nm. If the crystal has a half thickness of X nanometres, the Pb loss in percent will be approximately 15% * 200 / X. Recoil from smaller crystal faces will make the actual loss somewhat larger. More detailed examples are given in Davis and Davis, 2018.

## 5 Conclusions

The original aim of the experiment was to determine the Pb deletion profiles near natural faces of two baddeleyite crystals from separate samples but in both cases no such profile was observed, suggesting that the boundaries are cleavage related. Nevertheless, atom probe tomography of baddeleyite from the Great Dyke of Mauritania was used to constrain the average recoil distance from U decay to about 40 nm, based on the assumption that the observed U compositional gradient was due to internal zoning and the modified radiogenic Pb compositional gradient due to alpha recoil. A more precise estimation is
prevented by the limited size of the sample, which prevents knowledge of the composition over the entire range affected by recoil. The 40 nm estimate is significantly higher than that obtained from depth profiling of natural baddeleyite crystal surfaces using SHRIMP, but atom probe tomography of internal U zones appears to be less subject to systematic uncertainties than SHRIMP depth profiling of natural surfaces so, despite extrapolation uncertainty, we are inclined to have more confidence in the 40 nm estimate than the previous estimate of 24 ±7 nm (Davis and Davis, 2018). Further atom probe work on baddeleyite
to collect data adjacent to natural crystal surfaces and from U-zoned crystals over a one micron size range is necessary to improve the estimate of average alpha recoil distance.

This sample also shows evidence of primary U clusters with radii of a few nanometres. Depletion of radiogenic Pb within the clusters constrains alpha recoil distances to be more than a few nanometres. The clusters nevertheless show anomalously high Pb, although much less than for radioactive equilibrium, as well as high Fe and Ti concentrations, and U depletion around
their margins. Explaining their formation remains a challenge for crystal chemistry.

**Acknowledgements.**

Ulf Söderlund (Lund University) is thanked for supplying samples of baddeleyite from the Great Dyke of Mauritania.

**Code and data availability**. The code and supplementary data repository are found at:

https://osf.io/pxtjk/?view_only=65ec1411396b4a208634aae0f3f0341d






**Author contributions.**

SD conceived the project and furnished the samples. DF acquired and processed Atom Probe data. DWD wrote software to simulate recoil models and compare to data. All authors contributed to writing the manuscript.

**Competing interests.**

There are no competing interests.

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






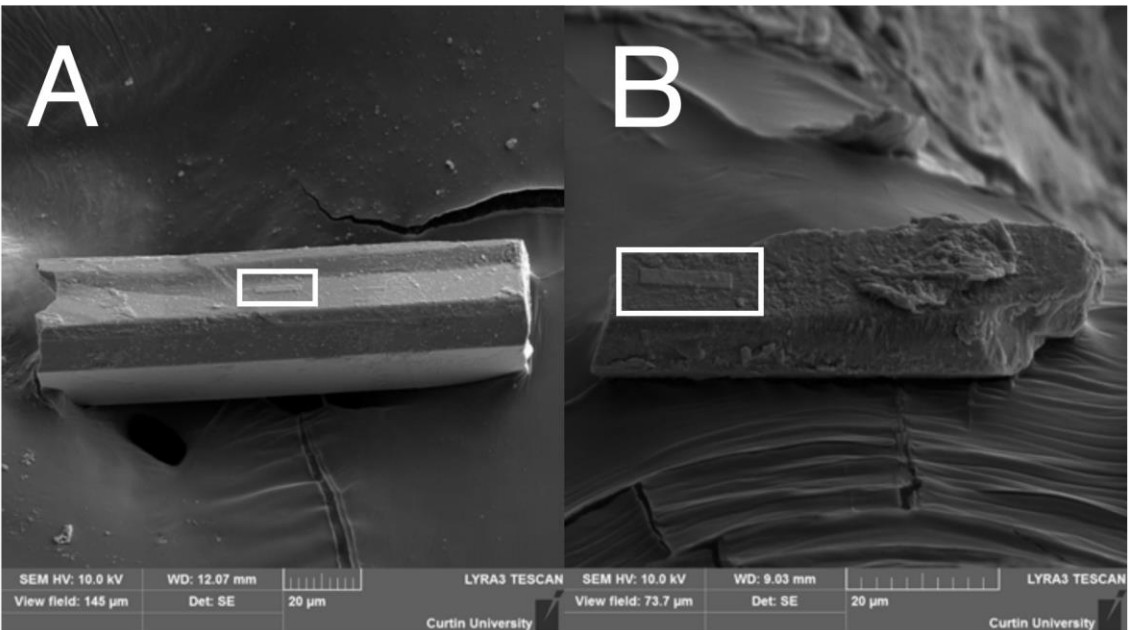

**Figure 1: Scanning electron photomicrographs of the two studied baddeleyite crystals: A = Hart Dolerite, B = Great Dyke of Mauritania. Rectangles denote apparent fresh crystal faces; higher-relief feature is the platinum strip applied to protect the selected region from gallium implantation and to facilitate handling of the selected area after being carved out by the focussed ion beam.**


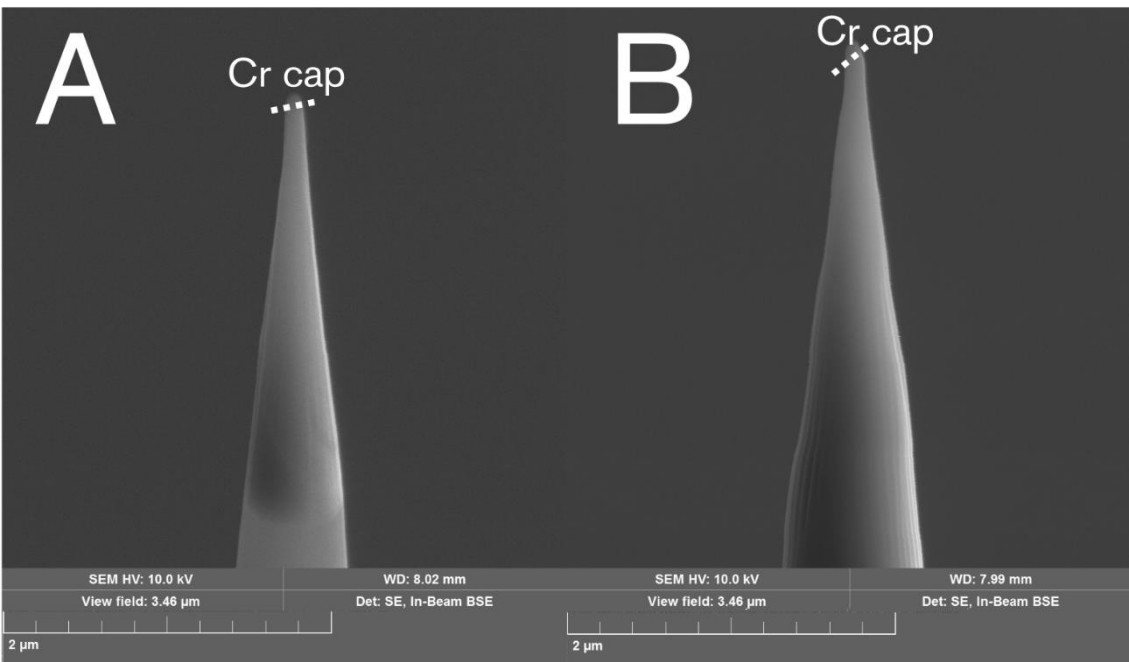

**Figure 2: Scanning electron micrograph photomicrographs of atom probe tomography specimens from (A) sample M5 of the Great Dyke of Mauritania and (B) sample M2 of the Hart Dolerite.**



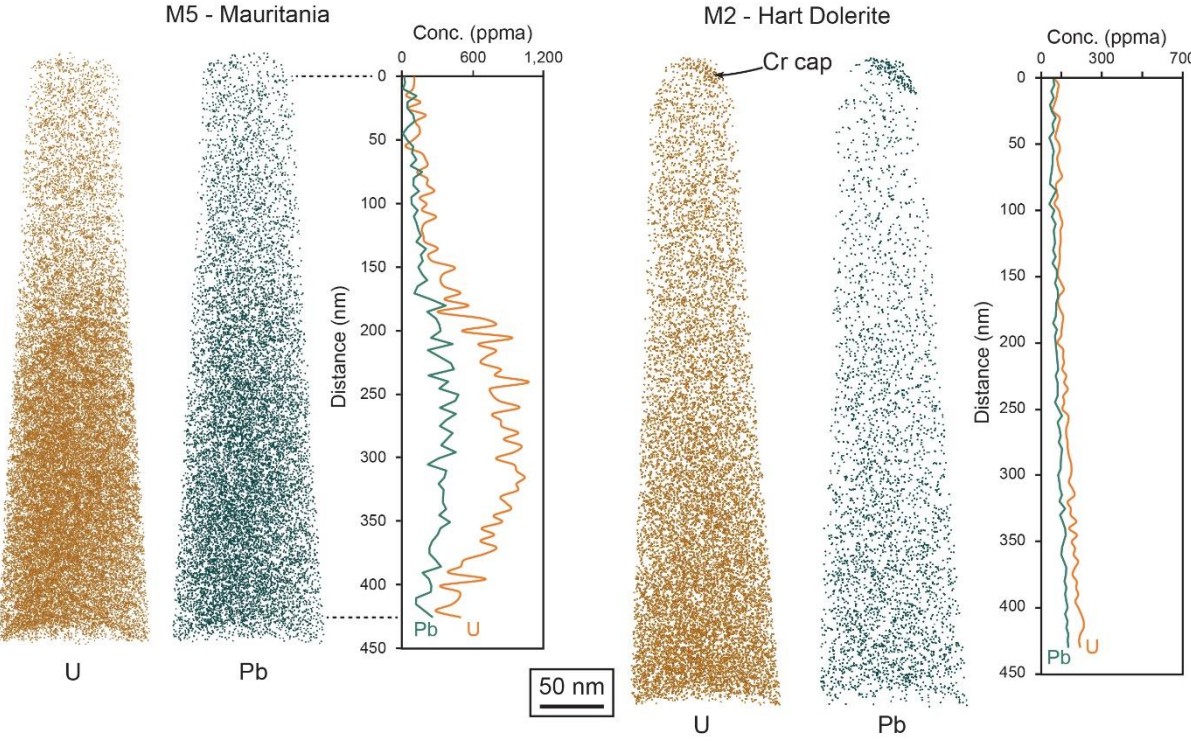

**Figure 3: Distribution of $^{238}$U and $^{206}$Pb atoms in samples M5 and M2 from atom probe tomography.**






**Figure 4: (A) measured ²³⁸U % concentration along the Z axis of sample M5. (B) Measured ²⁰⁶Pb/²³⁸U ratio along the Z axis of sample M5.**




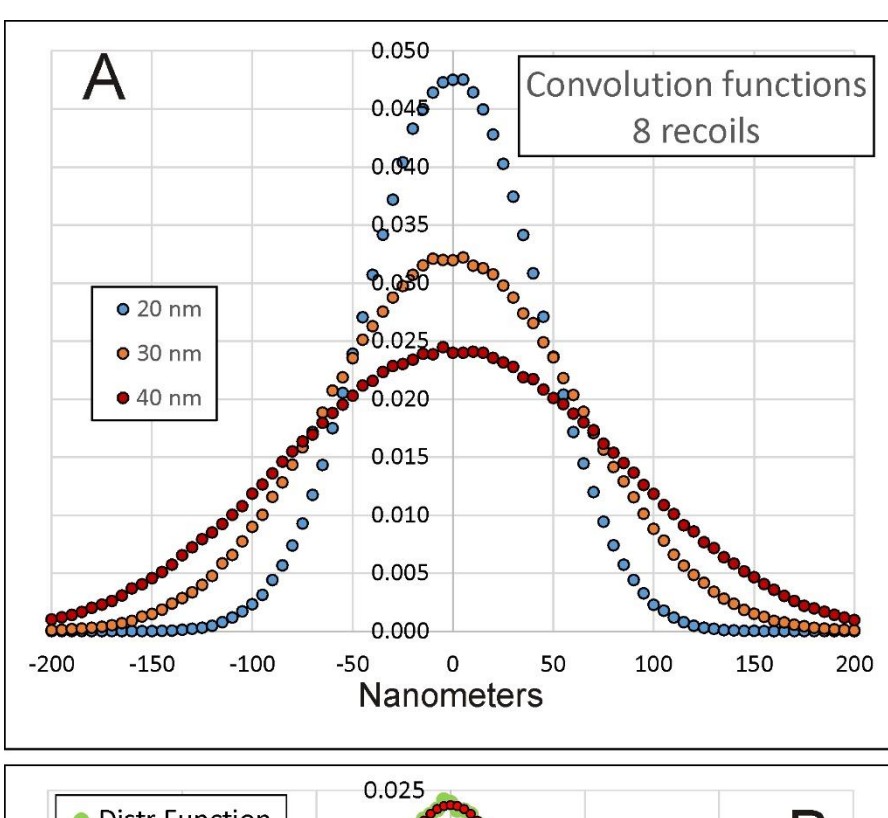

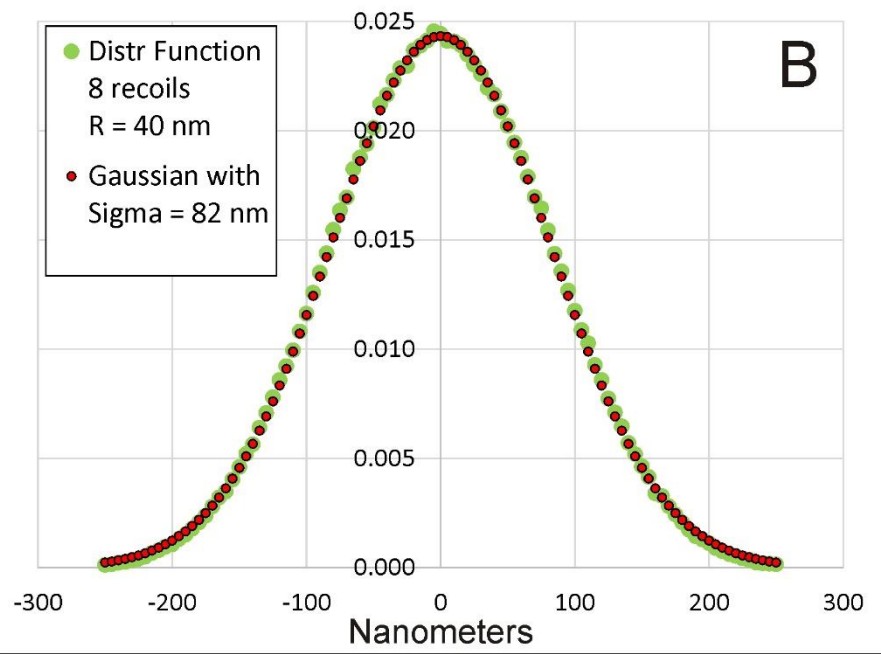

Figure 5: (A) Redistribution curves determined for 8 alpha recoils of 1,000,000 atoms with assumed recoil distances (R) of 20, 30 and 40 nm. The curves are normalized to an area of 1. (B) Comparison of redistribution curve for R = 40 nm with a Gaussian function with sigma of 82 nm.





Figure 6: (A) Hypothetical Gaussian shaped $^{238}$U distribution peak on a constant background and resulting $^{206}$Pb distribution after 8 alpha recoils with assumed recoil distance of 40 nm for each. (B) Resulting $^{206}$Pb/$^{238}$U ratio profile.






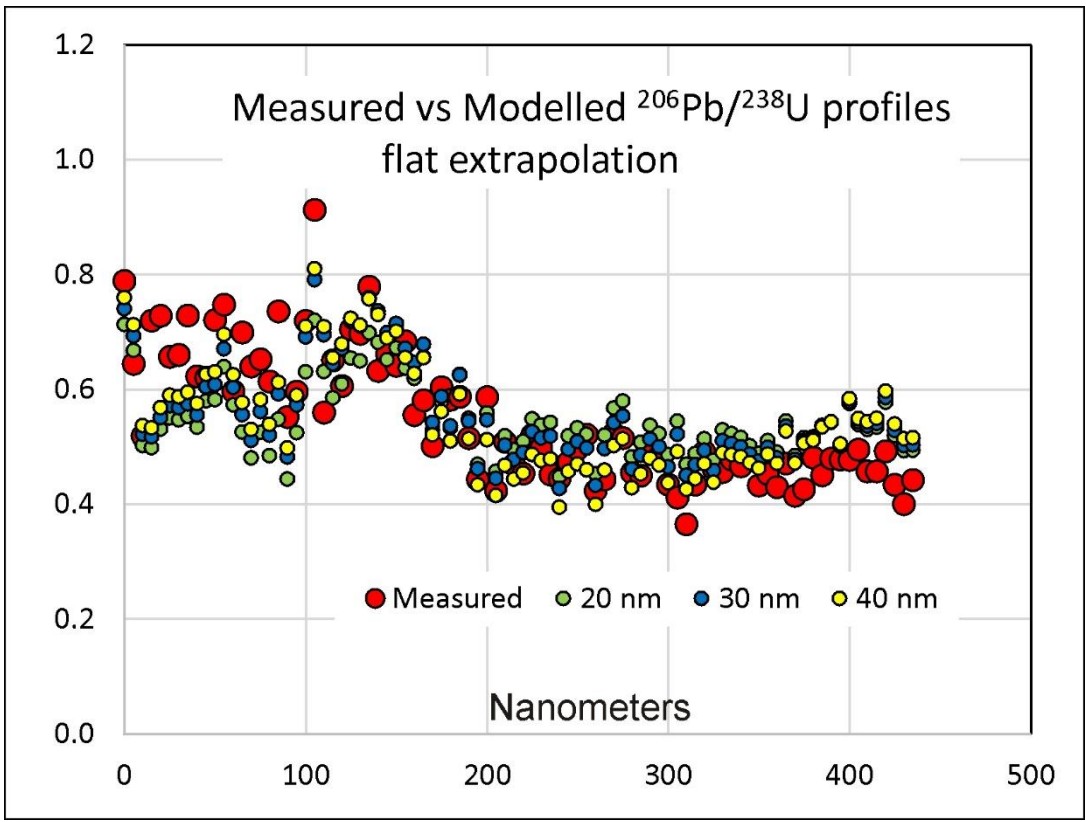

**Figure 7: Measured versus modelled $^{206}Pb/^{238}U$ profiles for different assumed values of average recoil distance across the measured distance range, assuming that the extrapolated U concentrations are constant and take the closest measured values above and below the measured distance range.**







**Figure 8: Measured versus modelled $^{206}$Pb/$^{238}$U profiles for an assumed value of average recoil distance of 40 nm with a single high-U zone.**



**Figure 9: Measured and extrapolated values in per cent for $^{238}$U concentration profile assuming uniform oscillatory zoning fitted to two parabolas.**





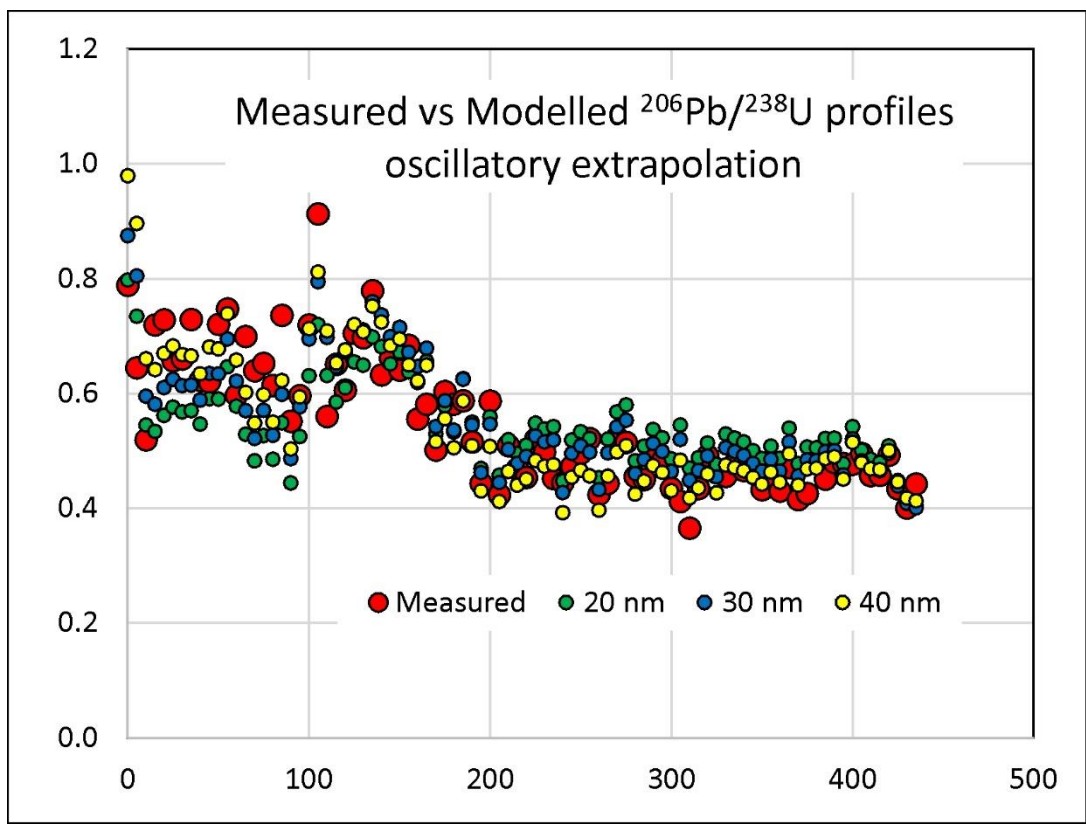

**Figure 10: Measured versus modelled $^{206}$Pb/$^{238}$U profiles for different assumed values of average recoil distance across the measured distance range, assuming that the extrapolated U concentrations follow the parabolic oscillatory pattern shown in Figure 9.**




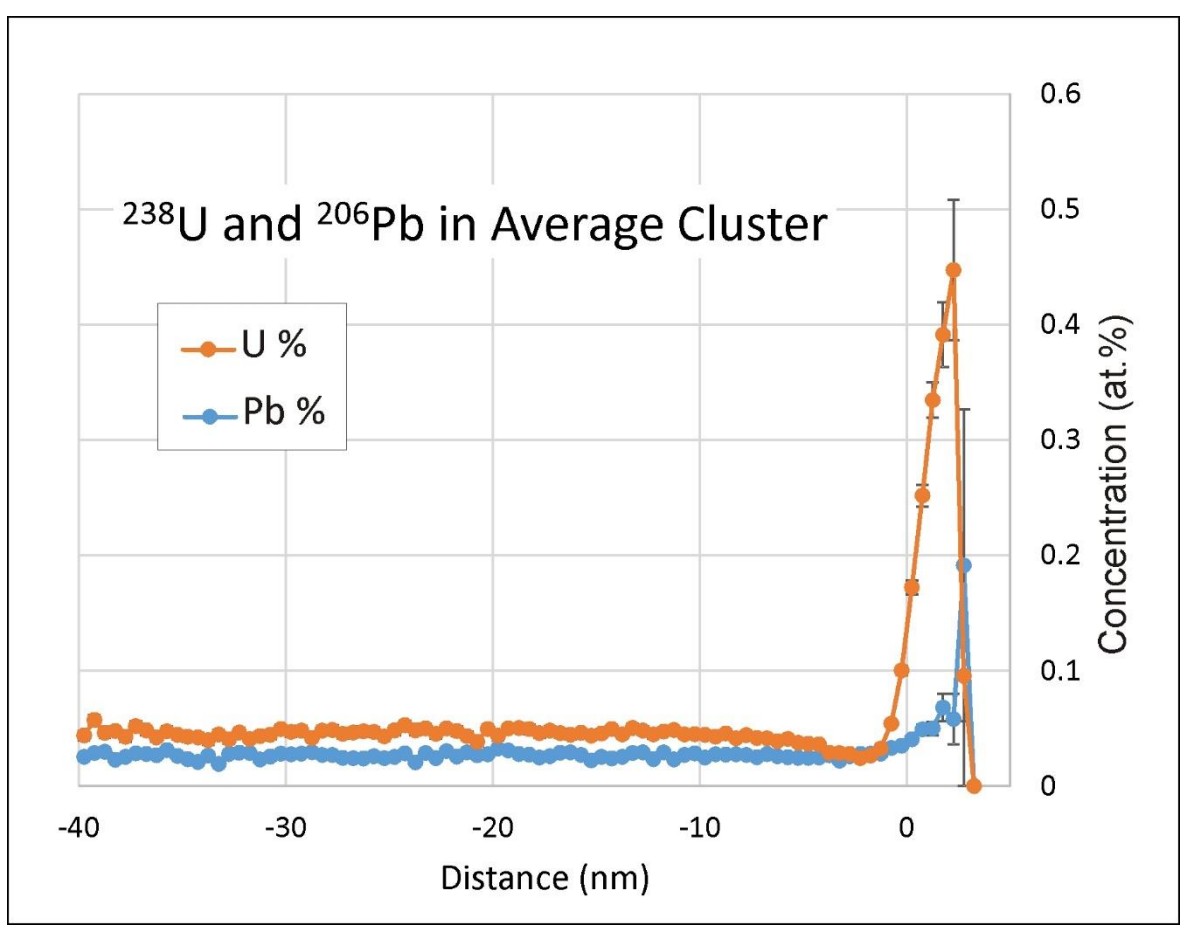

**Figure 11: Average $^{238}$U and $^{206}$Pb profile around U clusters in sample M5. The two U and Pb values at the highest positive distances (inside of cluster) are too imprecise to be meaningful.**





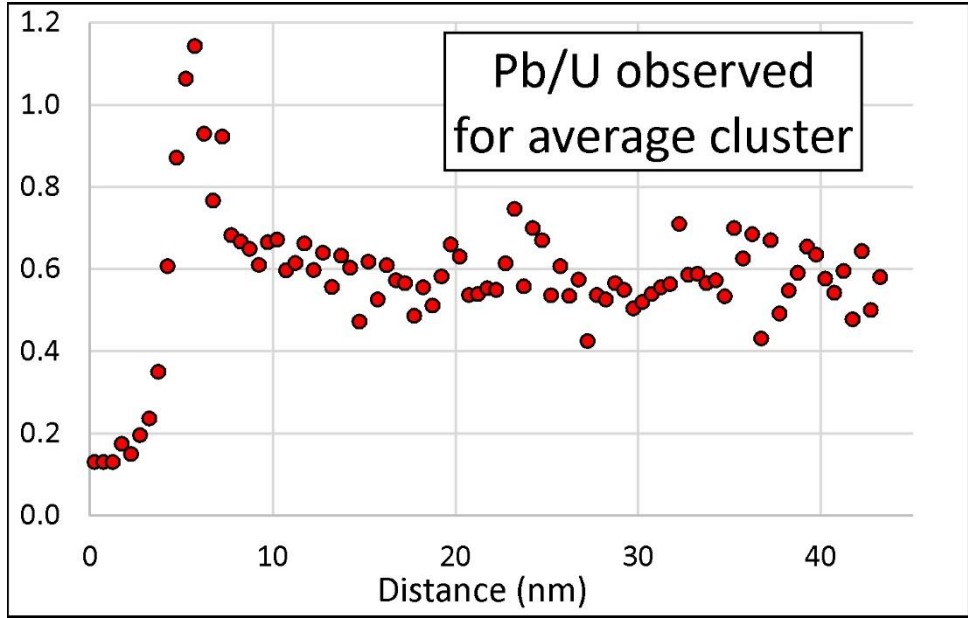


**Figure 12: Measured average $^{206}$Pb/$^{238}$U profile around U clusters in sample M5. Modelling with assumed average recoil (R) values above 1 nm produces similar profiles so clusters cannot be used to constrain realistic values of R.**







**Figure 13: Recoil profile from a uniform U distribution at a large baddeleyite crystal face (001) with assumed average recoil distances R of 0 nm (no redistribution) and 40 nm.**