# Peer review of "Constraints on average alpha recoil distance during 238U decay in baddeleyite (ZrO2) from atom probe tomography"

_Geochronology, 2023_

## Referee Comment (RC1)

This contribution attempts to quantify the average alpha recoil displacement of $^{206}$Pb in baddeleyite using atom probe tomography. Ejection of $^{206}$Pb from the rims of baddeleyite grains has the potential to produce U-Pb dates that are too young in small baddeleyite crystals with high surface area-to-volume ratios. If the average alpha recoil distance can be measured, Pb ejection corrections can be applied to U-Pb datasets to improve the accuracy of baddeleyite U-Pb geochronology. The study is well designed, and atom probe tomography has the appropriate spatial resolution to address this outstanding question. However, the presentation of the background, results, and discussion should be clarified and expanded (especially Section 4). Specific comments are listed below.

27: By what benchmark are concordant baddeleyite $^{206}$Pb/$^{238}$U dates younger than ~ 1000 Ma "too young"?

32: Radiation damage in zircon and baddeleyite cannot be deconvoluted into structure versus chemistry. The two are inherently linked. Baddeleyite is less susceptible to radiation damage because it typically incorporates less U than zircon, and oxides are more resistant to irradiation than silicates.

42: What is the closure temperature for Pb in baddeleyite? Is there Pb diffusion data available in the literature? Could younger baddeleyite U-Pb ages reflect Pb loss by volume diffusion?

60: How common is it to date baddeleyite crystals that are so small (10 - 15 μm) as to be affected by alpha recoil? Are larger baddeleyite grains not typically available?

Are there molecular dynamic simulation studies that have estimated alpha recoil distances in baddeleyite?

80: Are the quoted uncertainties 1 or 2 sigma?

124: Alpha recoils randomly redistributes U-daughters, not U.

126: Why would the largest U gradient be at the samples surface? Crystals can have all sorts of U-growth zonation. I assume you mean that recoil effects will be most apparent at the crystal surface, since recoil ejects a fraction of Pb atoms from the crystal causing a localized depletion in Pb relative to U.

134: What are the implications of no localized Pb depletion at the crystal surface if these are indeed crystal faces and not cleavage planes? This possibility should be explored further in the discussion before the authors move forward with their preferred interpretation. The SEM images of the Hart Dolerite crystal in particular really looks like a crystal face. Could you rotate the crystal and sample another face for atom probe?

173: Do not cite Wikipedia as a primary source.

194: the "modeled" $^{206}Pb/^{238}U$ profiles

198: Explain how you determined 40 nm to have the closest fit. Visual inspection or by some least squares parameter? What about recoil distances >40 nm?

201-204: I am confused by this sentence. Specify what you mean by observed profile. Do you mean projecting the end of the measured U profile downward improves the model fit to the measured $^{206}Pb/^{238}U$ profile?

205: While oscillatory growth zoning does commonly occur in some minerals, it is highly speculative to assume that this is the case here for a grain that is totally uncharacterized. Can the authors characterize the growth zoning in this grain or in a suite of other baddeleyite grains from the same sample?

225: Is the 40 nm value truly "robust" if models yielded high MSWD values? Nor can you really say that 40 nm is significantly higher than 24 ± 7 nm when you can't place uncertainties on your modeled value.

Section 3.2: It is not obvious to me from Figure 3 that there are U clusters in M5, and no visuals for Fe and Ti clusters are presented in the manuscript. Can the authors provide a Figures that show this more clearly?? At a minimum, the Fe and Ti data should be included since the data are referenced in the text.

Section 4: There are many methods for correcting (U-Th)/He ages for alpha ejection for different grain geometries, grain sizes, and surface-to-volume ratios. Some (U-Th)/He alpha ejection models even incorporate radionuclide zoning. The literature is fairly extensive on the topic. The discussion here could be expanded significantly by applying some of these methods to the case of Pb ejection in baddeleyite. It is important for the authors to demonstrate – given their preferred 40 nm estimate for alpha recoil – in what scenarios should geochronologist expect Pb ejection from baddeleyite grains to have a meaningful impact on U-Pb dates and how to correct them. Can recoil adequately explain the "too young" baddeleyite ages that the authors cite in the introduction?

Figure 2 & 3: It would be helpful to add some labels to the figures indicating where the crystal surface is for readers less familiar with atom probe tomography.

Figure 4: Label the x-axes. Adding a line at the expected equilibrium $^{206}Pb/^{238}U$ ratio (0.53) would be helpful.

Figure 5: Label the y-axes. Personally, I find having the y-axis labels in the center of the figure to be distracting.

Figure 6: Label the y-axis in A and x-axis in B.

Figure 10: Why not also show the 50 nm case, since it is discussed in the text?

Figure 11 & 12: Why is distance negative in Figure 11 but positive in Figure 12? It may be useful to demonstrate how different R values produce similar model results in Figure 12. I would be helpful to add a line at 0.53 for a better visual of the expected equilibrium value.

Supplement: I appreciate the author's total transparency in sharing all their modeling scenarios, however, including an active workbook with a hundred plots that don't all have labeled axes or enough context may be overkill. It the workbook is to be included, please label everything.

---

## Referee Comment (RC3)

Review Preprint Gchron-2023-15
Title: "Constraints on average alpha recoil distance during 238U decay in baddeleyite (ZrO2) from atom probe tomography"
Authors: Davis et al.

**General Comments**

This work highlights the use of Atom Probe Tomography (APT) to answer specific questions in geochronology and geochemistry, which are unattainable by more conventional techniques (e.g., SIMS, LA-ICP-MS, etc.). In this work, the authors focus on baddeleyite crystals, sourced from two localities dated previously using the high-precision U-Pb technique ID-TIMS. In both cases, the baddeleyite grains were reported to be discordant by up to 3%. The reason for discordance is suggested to be due to Pb loss from alpha recoil processes, thus resulting in Pb loss. The authors therefore use APT to analyze two baddeleyite grains, providing a single APT reconstruction from each respective sample. However, of the two samples, only one APT reconstruction exhibited U and Pb concentration profiles and a gradient in the $^{206}Pb/^{238}U$ ratio, which the authors suggest being a result of alpha recoil processes. The other sample showed no signs of disturbances, though concentrations of the relevant U and Pb isotope peaks were not detectable above background.

Although this work is relevant and needed, as baddeleyite is a widely used geochronometer in lithologies which do not saturate zircon, the data itself is quite sparse to support the quantity of related models and their interpretations presented in this manuscript. Their models for the calculation of alpha recoil are then technically based on one reconstruction – M5 (Ahmeyim Great Dyke). In contrast, M2 (Hart Dolerite) has a uniform U concentration and therefore the data from APT was inconclusive as to why these baddeleyite from the Hart Dolerite are discordant.

**Specific Comments**

My reserve with the manuscript as currently constructed is that it relies entirely on two APT reconstructions from two unique baddeleyite grains. Although they are quite large datasets for APT studies (65 and 62 million atoms), there is always the question of if the volumes analyzed are wholly representative of the system. Is there a reason more weren't analyzed? As I expect the authors will not analyze more data, I would suggest that the authors take more care into at least displaying more of the two reconstruction volumes (display more images, more angles, the U clustering, an isoconcentration surface if you truly find planar features in the volume…).

With only one APT reconstruction per sample, it is difficult to correlate what is observed to a very specific feature in the grain. Complimented by the general lack of corresponding techniques to rule out alternative options – e.g., these two grains could be mounted perpendicular to the FIB sections and imaged for CL at the least to view zonation and evidence for disturbances in the crystal lattice.

I find it interesting that the authors chose different locations with respect to crystallography from the two samples. In the Ahmeyim baddeleyite, they took a lift-out from the surface perpendicular to the C-axis, while in the Hart Dolerite baddeleyite they analyzed perpendicular to the A-axis(?). Could this contribute to the observed differences in concentration profiles relating to potential anisotropic differences in elemental diffusivities?

Although other studies have detailed the tedious process of extracting U and Pb isotopic concentrations from TOF spectra (e.g., Valley et al. 2014; Blum et al. 2018), I find it imperative that the TOF of these

two reconstructions are presented for reader evaluation of the runs as the entire study relies on the ability to resolve and quantify these two peaks in the mass spectra.

It's challenging to follow the discussion of alpha recoil relating to the concentration and ratio profile depicted in Figures 3 and 4, versus U and Pb clustering and the result of alpha recoil from enriched clusters of Uranium? There are no figures depicting this clustering, even though there is an entire discussion section dedicated to this topic: "3.2 Constraints on alpha recoil distance from U clustering".

Most significantly, their interpretation of the $^{206}Pb/^{238}U$ ratio profile as reflecting alpha recoil is opposite to the measured profile. Processes of alpha recoil at the crystal surface would result in the loss of Pb and result in a younger date (i.e., lower $^{206}Pb/^{238}U$ ratio) at the surface – while the measured profile indicates the opposite and instead progressively gets older toward the rim. It is possible that these measured profiles instead reflect a diffusive boundary, mirrored by the profile of U concentration. See Figure 10 of Ibanez-Mejia et al. (2014; Chemical Geology) for an example of this process. The authors should thus provide an explanation as to why their Pb compositional gradient could not be diffusion related and more thoroughly defend their interpretation of a gradient due to alpha recoil.

**Technical Corrections**

[line 20] It would be better if you could confirm these are indeed oscillatory patterns – e.g., image the grains analyzed or at least grains from these separates.

[line 24] A comma between lattice and but.

[line 31] How does baddeleyite break down into zircon if there's no supply of Si from the baddeleyite. I understand when zircon (ZrSiO4) breaks down into baddeleyite (ZrO2) and quartz (SiO2).

[line 83] What are the typical concentrations of U reported in these baddeleyite samples?

[line 84] Specify that these are ID-TIMS ages. Also, I read in the Ramsay et a. 2019 text that the Hart Dolerite Pb/Pb age is also an upper intercept age.

[line 94] How does the Cr cap ensure stable evaporation? I understood this would have the opposite effect…

[line 100] (mass spectrum in Dalton)

[line 104] Where are the TOF spectra for these two APT runs?

[line 112] "Lead was present as $^{206}Pb++$ and $^{207}Pb++$" – again we just have to take your word without the TOF spectra.

[line 116] Should the citations be ordered – either ascending or descending?

[line 120] I understand that APT is never the same as other methods, but it's interesting that the U is so low for the Hart Dolerite when the ID-TIMS gives U concentrations from 551 to 1682 ppm.

[line 127] "the largest U gradient should be encountered at the surface" – based on what? You could have oscillatory zones which have greater U concentration from earlier growth zones?

[line 153] "the planar symmetry of a zoned U distribution" - your reconstructions don't appear as having a plane of concentration change whereby indicating that this reflects a clear oscillatory zone/boundary? It's also challenging to see if this indeed is a boundary with only one view of the tip... the one chosen for Figure 3 is not particularly convincing.

[line 173] I'm sure you can find a source other than Wikipedia.

[line 235] The entire 3.2 constrains on alpha recoil distance from U clustering derives from U clustering which is never depicted in the figures?

[line 239] Do you have an explanation for why Ti of all elements is enriched in these clusters? No other elements?

[line 254] I think that Valley et al. 2014 and 2015 gave some explanations for clustering.

[line 262] You should also cite Peterman et al. 2019 for trace element enriched linear features.

[line 270] the clusters of U are primary, formed during initial crystallization: can you provide examples of this in the literature? Or explain this further?

[line 317] You confirm that you did not measure Pb depletion profiles yet go ahead and assume you can calculate alpha recoil from this profile? This Pb compositional gradient could be something other than alpha recoil: diffusion.

Figure 3: I suppose the concentration profiles are taken with respect to the observed volume? You should also plot the background levels with respect to each element analyzed here. What are the errors associated with each concentration point?

Figure 4: Where are the labels for each axis? Also the errors associated with these measurements??

Figure 7: Again what is the title for the y-axis? The points are incredibly difficult to see. I would suggest extending the graph to the full width of the page and spacing out the points so that you can see which correlate? Maybe use different symbols and not all circles?